# Hyposmia in COVID-19: Temporal Recovery of Smell: A Preliminary Study

**DOI:** 10.3390/medicina59091511

**Published:** 2023-08-22

**Authors:** Barbara Verro, Giulia Vivoli, Carmelo Saraniti

**Affiliations:** 1Division of Otorhinolaryngology, Department of Biomedicine, Neuroscience and Advanced Diagnostic, University of Palermo, 90127 Palermo, Italy; verrobarbara@gmail.com; 2Division of Plastic, Reconstructive, Microvascular and Aesthetic Surgery, Department of Maternal-Infant and Adult Surgical Medical Sciences, University of Modena and Reggio Emilia, 41121 Modena, Italy; giuliettaviv@gmail.com

**Keywords:** coronavirus, hyposmia, anosmia, olfactory perception, olfaction disorders

## Abstract

*Background and Objectives*: Hypo/anosmia is a characteristic symptom of COVID-19 infection. The aim of this study is to investigate the time of smell recovery and to identify a possible order of perception recovery of different odors in COVID-19 patients. *Materials and Methods*: A prospective observational study was conducted on not hospitalized COVID-19 patients, selected according to eligible criteria. The study was approved by the Ethical Committee. A questionnaire formulated by our team was submitted to patients in order to know the duration of the hypo/anosmia and hypo/ageusia and the order of odor recovery: vanillin (mixed olfactory/gustatory substances), phenyl ethyl alcohol (rosewater) (pure olfactory substances), eucalyptol (mixed olfactory/trigeminal substances), and eugenol (mixed olfactory/trigeminal/gustatory substances). *Results*: 181 patients were included. Hypo/ageusia and hypo/anosmia lasted on average 10.25 (±8.26) and 12.8 (±8.80) days, respectively. The most frequent odor recovery sequence was: (1) phenyl ethyl alcohol; (2) eucalyptol; (3) vanillin; and (4) eugenol. In COVID-19 patients, hypo/anosmia occurs more often in women and at a young age. *Conclusions*: This preliminary investigation highlighted novel data: there is a chronological order in perception recovery of different olfactory substances and, therefore, in the restoration of the various sensitive nerve pathways involved in the sense of smell.

## 1. Introduction

In December 2019, a new virus spread around the world causing more than 500 million cases of infection and more than 4 million cases of deaths: the 2019 coronavirus (COVID-19) [1,2]. On 11 March 2020, the World Health Organization (WHO) declared the state of a COVID-19 pandemic [3]. This virus belongs to Coronaviridae family that is characterized by a single-stranded RNA genome. The symptoms related to this infection are variable: there are asymptomatic forms, pauci-symptomatic forms (fever, myalgia, rhinorrhea, and headache) and severe, sometimes lethal, forms of respiratory failure [4]. In view of its predominant effect on the airways, this virus was defined and better known as severe acute respiratory syndrome coronavirus 2 (SARS-CoV-2). Moreover, studies demonstrated that smell and taste dysfunctions are frequently encountered symptoms of COVID-19 [5,6]. Because of this, several studies focused on COVID-19 hypo/anosmia in terms of symptom specificity since to date it is considered a warning and cardinal sign for COVID-19 infection [7,8]. Indeed, in April 2020, the Centers for Disease Control and Prevention (CDC) defined loss of smell and taste as one of the symptoms of COVID-19 [9]. In particular, this smell disorder usually appears within the first three days and represents the first symptom of COVID-19 infection in approximately 25% of cases [10,11,12] and more than 50% of infected patients reported smell impairment [13]. Some hypotheses have also been advanced on the pathogenesis of the symptom without any universally accepted results [14,15]. Therefore, with the studies having been conducted until this moment, we know that hypo/anosmia is a characteristic symptom of COVID-19 infection and that the smell recovery occurs on average in approximately 10 days [4,16]. However, little is known about the time and modes of olfactory recovery in COVID-19 patients.

Therefore, this study aims to investigate the recovery time of smell in COVID-19 patients and above all to detect any specific order of perception recovery for different odors. Second, the study aims to assess the time of recovery of hypo/ageusia too.

## 2. Materials and Methods

### 2.1. Patients’ Selection

A prospective observational study was carried out on patients who tested positive after a molecular swab for COVID-19, recruited in our city by general practitioners from 1 September to 31 December 2020. The study was approved by the Ethical Committee (approval number 11/2020). Informed consent was acquired verbally during the telephone interviews. The selection of patients was based on strict criteria. In particular, inclusion criteria were: (1) diagnosis of COVID-19 infection, laboratory-confirmed by real-time polymerase chain reaction (rt-PCR) on nasopharyngeal swabs, (2) smell impairment during COVID-19 infection, (3) mild to moderate COVID-19 infection that does not require hospitalization; (4) over 18 years of age, (5) ability to understand and will to answer the questionnaire, and (6) Italian native speakers. Exclusion criteria were: (1) pre-existing hyposmia or anosmia, (2) presence of comorbidities (infectious, sinonasal disease and/or surgery, neurological or psychiatric diseases, allergic rhinitis, previous head trauma, other) and/or drug use that could be responsible for hyposmia or anosmia, (3) previous head and neck chemotherapy or radiotherapy, (4) pregnancy, (5) patients hospitalized for COVID-19, (6) inability to find the substances required to participate in the survey, and (7) reported allergy to any of the substances necessary for the study.

### 2.2. Study Protocol

The recruited patients were contacted by telephone several times during the quarantine. The first telephone interview took place on the same day as the notification of the certified positive molecular swab for COVID-19 infection. During the interview, after acquiring the patient’s consent, the first part of the questionnaire was administered in order to select the patients who met the selection criteria. The questionnaire was formulated by our team and always administered by the same otolaryngologist; it consists of 9 items and can be divided into two parts (Figure 1).

In the first part, the following data are collected: age (year), gender, possible comorbidities (diabetes mellitus, infections, sinonasal disease and/or surgery, neurological disease, allergic rhinitis), previous head and neck radio-chemotherapy, and/or drug use that could be responsible for hyposmia or anosmia. Then, our team asked the patients to find and smell some common olfactory substances: 4-hydroxy-3-methoxybenzaldehyde (or vanillin), phenyl ethyl alcohol (rosewater), eucalyptol, and eugenol [17]. The patients were contacted by telephone weekly in order to progressively complete the second part of the questionnaire. Patients who did not report hypo/anosmia during quarantine (at least three weeks until the COVID-19 nasopharyngeal swab was negative) were excluded from the study. In the second part of the questionnaire, the symptoms developed due to the COVID-19 infection were examined: rhinorrhea, nasal obstruction, sore throat, fever, reduced or loss of smell (hyposmia or anosmia, respectively), and reduced or loss of taste (hypogeusia or ageusia, respectively). In particular, with the questionnaire, we focused on the duration of hypo/anosmia and in which order the various olfactory substances previously asked to the patients (4-hydroxy-3-methoxybenzaldehyde or vanillin, phenyl ethyl alcohol or rosewater, eucalyptol, and eugenol) were recovered. In fact, each of these olfactory substances belongs to a different class of substances able to stimulate different sensitive neural pathways: pure olfactory (phenyl ethyl alcohol), olfactory-trigeminal (eucalyptol), olfactory-gustatory (4-hydroxy-3-methoxybenzaldehyde or vanillin), and olfactory-gustatory-trigeminal (eugenol) pathway [18] (Table 1). The last item of the questionnaire concerned the duration of hypo/ageusia (days). Based on the answers obtained, the average duration of “hypo/anosmia” and “hypo/ageusia” symptoms and the odor recovery sequence were calculated.

The patient was instructed to smell single olfactory substances once a day, preferably in the morning, and to note the level of smell: absence of smell sensitivity, poor smell sensitivity, or complete recovery of smell. The patient was contacted weekly to report on the progress of the eventual smell recovery.

### 2.3. Statistical Analysis

Duration of symptoms “hypo/anosmia” and “hypo/ageusia” was expressed in days. Categorical variables (sex, age, COVID-19 symptoms, and possible drugs) and results were reported as numbers, percentages of the total and/or mean ± standard deviation (SD). Descriptive statistics were used.

## 3. Results

The process selection of patients is shown in Figure 2.

The number of patients identified by general practitioners who had joined the study was 327. In total, 112 patients were excluded based on the selection criteria, after the administration of the first part of the questionnaire to patients. During the evaluation, another 34 patients were excluded because they had not experimented hypo/anosmia during COVID-19 infection. Therefore, only 181 patients were included in the study (Table 2).

Among the enrolled patients, 59.67% (108) were female and the average age was 32.34 (±11.09) years (range 18–66). Moreover, the following COVID-19-related symptoms were evaluated: rhinorrhea (61.88%), nasal obstruction (43.64%), sore throat (26.86%), fever (65.74%), and hypo/ageusia. Most patients (170/181) reported hypo/ageusia (93.92%) with taste recovery in 10.25 (±8.26) days on average (range 2–30). In particular, 96 patients reported taste recovery within the first week (1–7 days), 37 patients during the second week (8–14 days), 19 patients in the third week (15–21 days), only 5 patients in the fourth week (22–28 days), and 13 patients within the fifth week (29–34 days) (Figure 3).

Other very frequent symptoms reported by patients were fever (65.74%) and rhinorrhea (61.88%). Sore throat was reported by only 26.86% of patients and nasal obstruction by 43.64% of patients. Furthermore, 94.47% of patients did not take any therapy to treat these symptoms, 5.53% used nasal decongestants, and no patients took oral corticosteroids. The recovery time interval for the smell was variable, from 3 to 30 days, with an average of 12.8 (±8.80) days. In particular, as shown in Figure 4, 73 patients reported smell recovery during the first week (1–7 days), 52 patients within the second week (8–14 days), 28 patients in the third week (15–21 days), only 6 patients during the fourth week (22–28 days), and 22 patients in the fifth week (29–34 days).

The order of perception recovery for the four examined substances (phenyl ethyl alcohol, eucalyptol, vanillin, and eugenol) provides 24 different possible combinations, according to the formula of permutations without repetition of elements (Pn: n!) (Table 3).

As can be seen in Table 3 and Figure 5, in our case history we found only 5 combinations among the 24 possible ones:Sequence #1 (62.43%): phenyl ethyl alcohol, eucalyptol, vanillin, eugenol;Sequence #2 (9.94%): phenyl ethyl alcohol, vanillin, eucalyptol, eugenol;Sequence #3 (12.15%): eucalyptol, phenyl ethyl alcohol, vanillin, eugenol;Sequence #4 (9.40%): vanillin, phenyl ethyl alcohol, eucalyptol, eugenol;Sequence #5 (6.08%): vanillin, eucalyptol, phenyl ethyl alcohol, eugenol.

The perception of pure olfactory substances (phenyl ethyl alcohol) was first recovered in most patients (72.37%). In particular, the most frequently encountered odor recovery sequence was “sequence #1”, that is: (1) phenyl ethyl alcohol; (2) eucalyptol; (3) vanillin; and (4) eugenol.

In particular, analyzing the results of our survey, we found that phenyl ethyl alcohol was resumed in 13 ± 8.43 days on average, eucalyptol in 13.87 ± 8.71 days on average, vanillin in 14.43 ± 9.04 days on average, and eugenol in 16.16 ± 9.02 days on average.

## 4. Discussion

Several studies were published regarding COVID-19 infection and its clinical manifestations, including severe ones, in the world population. In 2020, two meta-analysis demonstrated that olfactory and gustatory disfunctions are quite common in COVID-19 infection, in 41% and 62% of cases, respectively [19,20]. A study by Lechien et al. [5] also reported a statistically significant positive association between the two symptoms. The same result was found and confirmed in other studies [4,16,21] as well as in our sample in which 93.92% of patients reported both hypo/anosmia and hypo/ageusia. Most of the patients of our cohort were females (59.67%) and the mean age was 32.24 (±11.09) years. Similar epidemiological results are reported in the literature [5,14,16,22] with an average age between 36 and 52 years. By the way, recent studies found that COVID-19 smell disorders are more common in females than in males, probably due to different steroid levels, different immunological system and, most of all, due to different expression of Angiotensin-Converting-Enzyme-2 (ACE2) that, as explained later, seems to be involved in hyposmia [23,24,25]. Regarding the age category, some studies hypothesized that middle-aged people have the highest levels of ACE2 receptors [26,27,28]. A recent study also demonstrated that recovery of smell may depend on severity of olfactory impairment, with poorer prognosis in older and anosmic patients [29].

COVID-19 infection often causes other symptoms as well: rhinorrhea, fever, nasal obstruction, sore throat, and headache [30,31,32]. The role of nasal obstruction and/or rhinorrhea in the pathogenesis of hypo/anosmia was studied, excluding any statistically significant association [5,16,33]. Moreover, a study found that smell disorder is more common in asymptomatic COVID-19 patients [34]. This result led us to hypothesize that other pathogenetic mechanisms could be involved in smell impairment, also considering that viral rhinitis and nasal congestion cause hypo/anosmia that resolves in a few days. Based on the lack of link between nasal congestion and loss of smell, Mastrangelo et al., suggested that COVID-19 may cause sensorineural damage (olfactory sensory neurons) [35]. However, a review by Butowt et al. ruled out this hypothesis because of three data that are inconsistent with this thesis: (1) the olfactory receptor neurons regeneration requires approximately 10 days, which is more than the minimum time of smell recovery (3 days); (2) the viral entry proteins (ACE2) are not expressed by olfactory receptors neurons and so these cells cannot be infected by SARS-CoV-2; (3) several studies demonstrated the presence of the virus in the sustentacular cells but not in the olfactory receptors neurons. Another hypothesis of COVID-19 smell dysfunction was the infiltration of the brain through olfactory receptor neurons; however, as written above, these cells do not express virus entry proteins and therefore SARS-CoV-2 cannot reach the brain through the anterograde axonal transport [8]. Thus, studies have hypothesized that hypo/anosmia may be caused by the entrance of COVID-19 into sustentacular and horizontal basal cells thanks to ACE2 [33,36,37]. These non-neural cells are involved in the maintenance and renewal of the olfactory mucosa and their involvement could therefore be responsible for an inflammatory response with consequent impairment of olfactory sensory neurons [8,14,38,39]. In favor of this hypothesis there are three data: (1) these support cells express high levels of viral entry proteins, such as ACE2; (2) the sustentacular cells regeneration is faster than olfactory receptor neurons, so this can explain the rapid smell recovery; (3) the high expression levels of viral entry proteins lead to higher viral loads in the nose than in other sites of respiratory tract, proving why anosmia usually represent the first symptom of SARS-CoV-2 infection. According to this 2021 review, COVID-19 smell impairment may be caused by two possible ways: (a) direct damage of sustentacular cells that cannot perform their function on the olfactory epithelium; (b) indirect effect, that is the damage of olfactory receptor neurons due to lack of cell protection by sustentacular cells [8].

Furthermore, the frequent finding of hypo/anosmia in COVID-19 patients with mild to moderate symptoms led us to think that the virus may spread through two paths: nasal with mild to moderate symptoms with hypo/anosmia and pulmonary with a high risk of respiratory failure [19,40].

The study then evaluated the duration of hypo/anosmia and hypo/ageusia revealing that patients recovered smell and taste in 12.8 (±8.80) and 10.25 (±8.26) days on average, respectively, consistent with the data reported in the literature [4,5,16,41]. This result confirms the hypothesis that hypo/anosmia and hypo/ageusia are not caused by common viral rhinitis. With regards to the taste impairment, there are few studies and theories. In particular, based on the same assumptions of smell dysfunction, viral entry proteins (ACE2) have been searched and found on taste cells, tongue cells, and gingival tissue, explaining the direct damage of these cells. Another hypothesis stated an indirect effect of the virus on taste cells: in fact, SARS-CoV-2 induces inflammatory cytokines that cause gustatory nerves damage [42]. However, studies reported a difficulty in understanding whether taste dysfunction is related to smell impairment or whether it is a standalone symptom [21,43].

Compared to the current literature, the novel data that we analyzed concerns the time course of recovery of the perception of different types of odors. In fact, the olfactory substances we have evaluated are able to stimulate one or more nervous sensory systems: olfactory, trigeminal, and gustatory. In total, 113 patients (62.43%) of our case history reported the following recovery order in the perception of the evaluated olfactory substances: pure olfactory substances (phenyl ethyl alcohol), mixed olfactory/trigeminal substances (eucalyptol), mixed olfactory/gustatory substances (vanillin), and mixed olfactory/trigeminal/gustatory substances (eugenol). It is quite clear that the perception of mixed olfactory/trigeminal/gustatory substances is recovered lastly since the three different sensory pathways are involved. In the same way, the early recovery of perception of pure olfactory substances is understandable, due to the involvement of only one sensory path. The trigeminal system usually allows the perception of burning, freshness, tickling, and/or stinging of the olfactory substances (e.g., eucalyptol) following the stimulation of the fibers of the trigeminal nerve [44]. The gustatory system, on the other hand, allows one to differentiate sweet, bitter, salty, sour, and umami through nerve fibers of three cranial nerves: facial, glossopharyngeal, and vagus [45]. These premises lead us to hypothesize that the involvement of multiple first-order neurons and the complexity of the gustatory system functioning could explain the late recovery in the perception of mixed olfactory/gustatory substances (e.g., vanillin). More studies are therefore needed to understand the underlying pathogenetic mechanism of this chronological order of recovery of the smell in COVID-19 infection.

However, the present study has some limits. First, no distinction was made between the symptoms “hyposmia” and “anosmia”, as they were considered as a single category; this could affect the assessment of symptom duration. In order to obviate this, we have expressed the “duration” value as mean ± SD. Furthermore, in order to avoid the distortion of this parameter evaluation by any local or systemic therapy, we asked the patients about their current treatments during the evaluation period. The survey showed that no patients had taken oral steroids. The use of nasal decongestants has no influence on the recovery of the smell, considering that nasal obstruction and rhinorrhea were not significantly associated with hypo/anosmia. Second, the evaluation of hypo/anosmia and the subsequent recovery of the sense of smell were evaluated only through a questionnaire and without the help of any specific and objective instrumental tests, which were not performable because patients were in quarantine for the high risk of COVID-19 contagion. Indeed, the study refers to the first wave of COVID-19 infection during which it was very contagious, with high mortality and a lack of vaccines. In any case, our questionnaire was formulated on the basis of other questionnaires already present in the literature [44] and considered valid tools for obtaining information on olfactory and gustatory disorders in COVID-19 patients [21,22,46,47,48,49]. Furthermore, the questionnaire was always administered by the same otolaryngologist to avoid bias due to different examiners who could explain the questions differently. Third, our sample included only patients with mild-to-moderate COVID-19 symptoms who were not hospitalized. Hence, it may not be representative of the general population; however, it would not have been possible or ethical to administer the questionnaire to patients in serious clinical conditions. By the way, many studies reported that smell and/or taste impairments resulted more commonly in non-hospitalized patients than in hospitalized ones [50,51]. Fourth, our study’s sample size is relatively small compared to the worldwide spread of the infection. So, other studies with larger samples would be needed in order to confirm our preliminary results and to investigate the underlying pathogenetic mechanisms of olfaction recovery according to a precise sequence of reactivation of the various sensory nerve pathways involved.

## 5. Conclusions

Hypo/anosmia is now widely recognized as a warning sign of COVID-19 infection thanks to the various studies carried out in 2020. Based on these results, our study observed and analyzed the main characteristics of gustatory and olfactory impairment and on the subsequent recovery of taste and smell in COVID-19 patients. Moreover, the present study highlighted novel data: there is a chronological order in the perception recovery of the different olfactory substances and therefore in the restoration of the various sensitive nerve pathways involved in smell. However, this is a preliminary study and future studies need to elucidate the pathogenetic mechanisms underlying this method of smell recovery in COVID-19 infection.

## Figures and Tables

**Figure 1 medicina-59-01511-f001:**
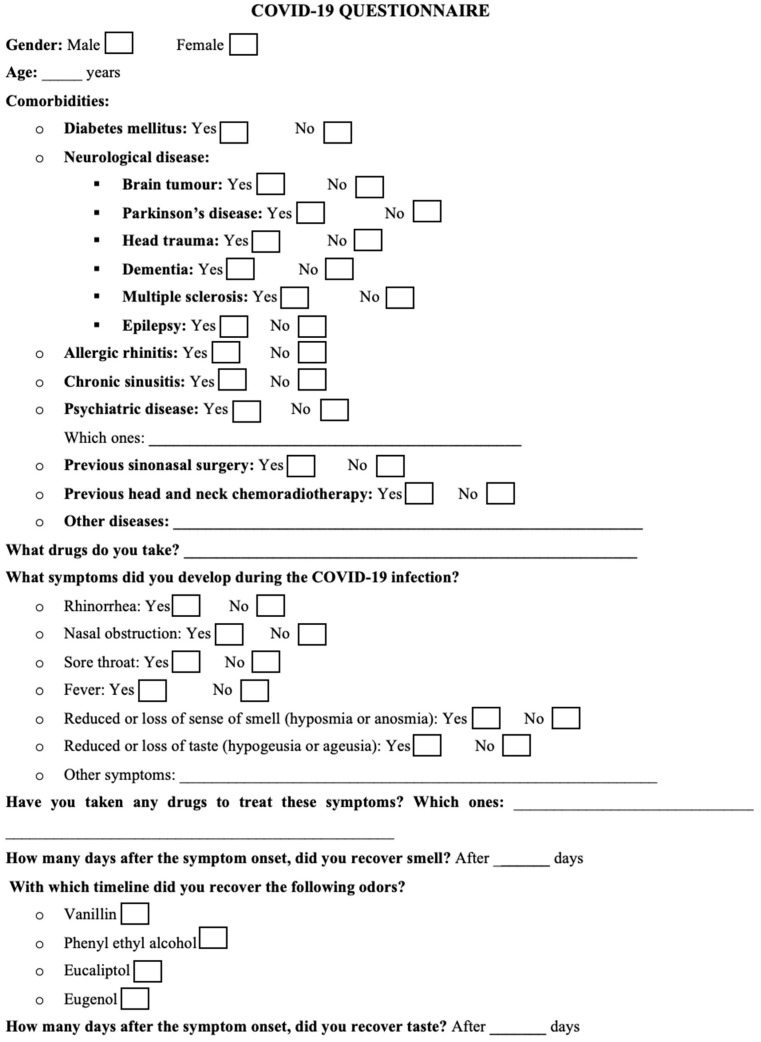
Questionnaire for COVID-19 patients.

**Figure 2 medicina-59-01511-f002:**
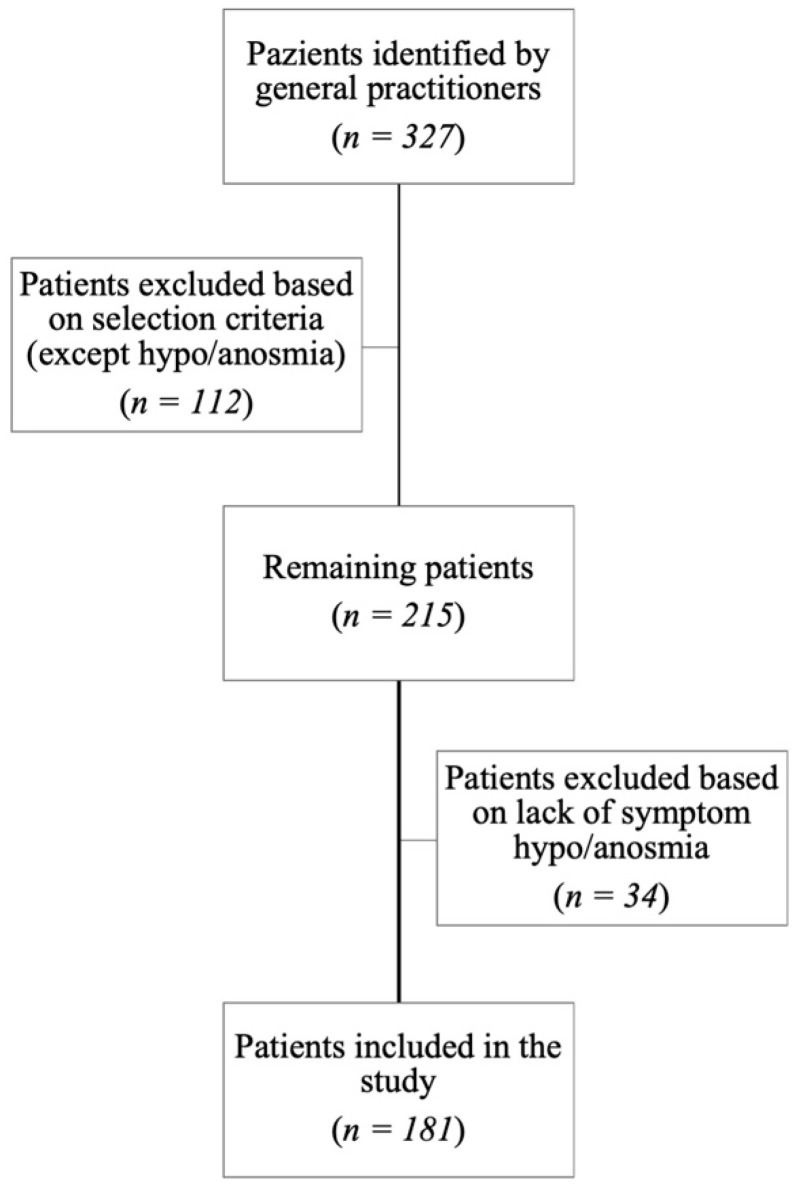
Patients’ selection.

**Figure 3 medicina-59-01511-f003:**
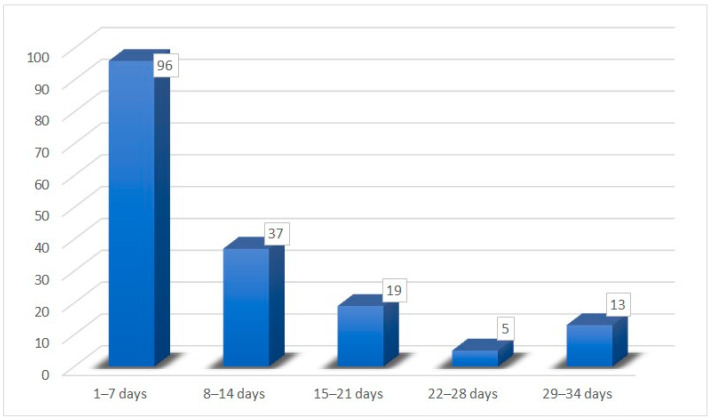
Recovery time for hypo/ageusia.

**Figure 4 medicina-59-01511-f004:**
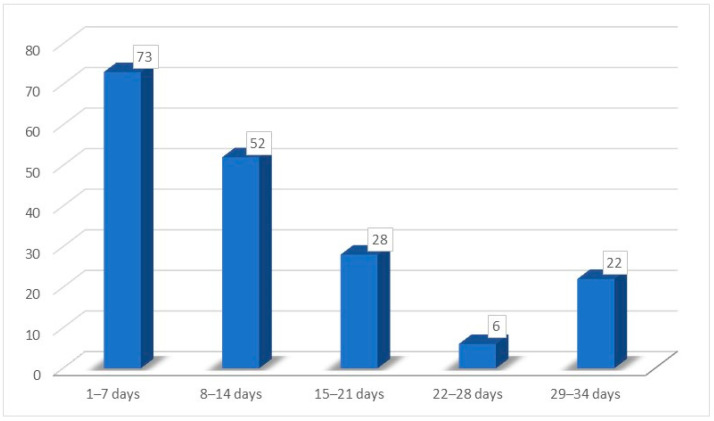
Recovery time for hypo/anosmia.

**Figure 5 medicina-59-01511-f005:**
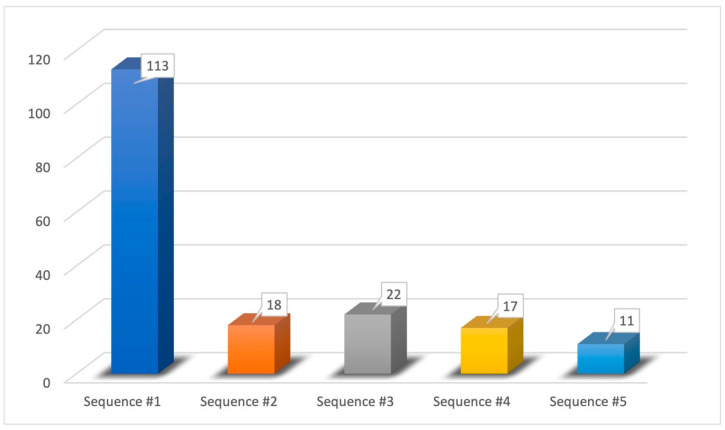
Order of odors perception recovery [sequence #1 (phenyl ethyl alcohol, eucalyptol, vanillin, eugenol), sequence #2 (phenyl ethyl alcohol, vanillin, eucalyptol, eugenol), sequence #3 (eucalyptol, phenyl ethyl alcohol, vanillin, eugenol), sequence #4 (vanillin, phenyl ethyl alcohol, eucalyptol, eugenol), and sequence #5 (vanillin, eucalyptol, phenyl ethyl alcohol, eugenol)].

**Table 1 medicina-59-01511-t001:** Types of olfactory substances.

Types of Olfactory Substances	
Pure olfactory substances	Phenyl ethyl alcohol
Mixed olfactory/trigeminal substances	Eucalyptol
Mixed olfactory/gustatory substances	Vanillin
Mixed olfactory/trigeminal/gustatory substances	Eugenol

**Table 2 medicina-59-01511-t002:** Characteristics of enrolled patients.

Parameters	N° Patients (%)
**Sex**	
Female	108 (59.67)
Male	73 (40.33)
**Age**	
Mean age	32.34 (±11.09)
Minimum	18
Maximum	66
**COVID-19 ENT symptoms**	
Rhinorrhea	112 (61.88)
Nasal obstruction	79 (43.64)
Sore throat	45 (26.86)
Fever	119 (65.74)
Hypo/ageusia	170 (93.92)
**Drugs taken for COVID-19 infection**	
None	171 (94.47)
Nasal decongestants	10 (5.53)
Oral corticosteroids	0
**Recovery time for hypo/anosmia**	
Mean time	12.8 (±8.80) days
Minimum time	3 days
Maximum time	30 days
**Recovery time for hypo/ageusia**	
Mean time	10.25 (±8.26) days
Minimum time	2 days
Maximum time	30 days
**Total**	181 (100)

**Table 3 medicina-59-01511-t003:** Possible combination of time course of smell perception recovery.

Combination	Sequence of Smells Perception Recovery	N° Patients (%)
1	Phenyl ethyl alcohol → eucalyptol → vanillin → eugenol	113 (62.43)
2	Phenyl ethyl alcohol → vanillin → eucalyptol → eugenol	18 (9.94)
3	Eucalyptol → phenyl ethyl alcohol → vanillin → eugenol	22 (12.15)
4	Vanillin → phenyl ethyl alcohol → eucalyptol → eugenol	17 (9.40)
5	Vanillin → eucalyptol → phenyl ethyl alcohol → eugenol	11 (6.08)
6	Phenyl ethyl alcohol → eucalyptol → eugenol s → vanillin	0
7	Phenyl ethyl alcohol → vanillin → eugenol → eucalyptol	0
8	Phenyl ethyl alcohol → eugenol → eucalyptol → vanillin	0
9	Phenyl ethyl alcohol → eugenol → vanillin → eucalyptol	0
10	Eucalyptol → phenyl ethyl alcohol → eugenol → vanillin	0
11	Eucalyptol → vanillin → phenyl ethyl alcohol → eugenol	0
12	Eucalyptol → vanillin → eugenol → phenyl ethyl alcohol	0
13	Eucalyptol → eugenol → vanillin → phenyl ethyl alcohol	0
14	Eucalyptol → eugenol → phenyl ethyl alcohol → vanillin	0
15	Vanillin → phenyl ethyl alcohol → eugenol → eucalyptol	0
16	Vanillin → eucalyptol → eugenol → phenyl ethyl alcohol	0
17	Vanillin → eugenol → eucalyptol → phenyl ethyl alcohol	0
18	Vanillin → eugenol → phenyl ethyl alcohol → eucalyptol	0
19	Eugenol → phenyl ethyl alcohol → eucalyptol → vanillin	0
20	Eugenol → phenyl ethyl alcohol → vanillin → eucalyptol	0
21	Eugenol → vanillin → phenyl ethyl alcohol → eucalyptol	0
22	Eugenol → vanillin → eucalyptol → phenyl ethyl alcohol	0
23	Eugenol → eucalyptol → vanillin → phenyl ethyl alcohol	0
24	Eugenol → eucalyptol → phenyl ethyl alcohol → vanillin	0
**Total**		**181 (100)**

## Data Availability

Not applicable.

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
