# Peer review of "Hyposmia in COVID-19: Temporal Recovery of Smell: A Preliminary Study"

_medicina, 2023, doi:10.3390/medicina59091511_

Round 1
Reviewer 1 Report
Thanks for the opportunity to review the article titled: "Hyposmia in Covid-19: temporal recovery of the different nerve pathways. A preliminary study." The study investigated the time of smell recovery and the order of perception recovery for different odors in COVID-19 patients.
The abstract clearly states the background, objectives, methods, results, and conclusions of the research. The abstract effectively communicates the aim of the study, the number of participants, the duration of hypo/ageusia and hypo/anosmia, and the main findings regarding the chronological order of perception recovery.
The introduction provides a comprehensive overview of COVID-19, its symptoms, and the significance of olfactory dysfunction in COVID-19 infection. It mentions previous studies on COVID-19 hypo/anosmia, highlights the prevalence of these symptoms, and discusses possible pathogenesis. The introduction sets a clear context for the study, citing relevant literature, and identifies the research gap regarding the recovery time for smell in COVID-19 patients.
The methods section described the study design, patient selection criteria, and data collection procedures. However, it lacks information on sample size determination and statistical analyses beyond descriptive statistics.
The discussion section interprets the study findings in the context of existing literature and highlights the prevalence of hypo/anosmia in COVID-19 patients and confirms previous observations. The limitations of the study are acknowledged, such as the absence of objective tests and the small sample size. The discussion also provides insights into future research directions.
Author Response
Thanks for the comments.
Unfortunately, due to the design of the study, it was not possible to make statistical studies about the possible link between the various data. However, it's a preliminary study and we hope that it will be an incentive for further research.
Reviewer 2 Report
Medicina 2531503: Hyposmia in COVID-19: temporal recovery of the different nerve pathways. A preliminary study.
This is yet another paper on the symptoms by COVID-19 focusing on olfactory dysfunction. Although there are so many other papers already published on the similar topic, I started reading the paper with expectation of some new information on the recovery of chemical senses depending on the type of smells. Unfortunately, I encountered some basic problems in the methods that I cannot approve the publication of the paper. Especially the grouping of the types of odors is a problem. The authors claim that they followed the classification of a study (reference 14), but the study of reference 14 does not classify odors in 4 groups. Also, the authors list several odors they used to represent each type of odors, and some of them are single chemical compounds whereas some are odors that contain hundreds of different chemical compounds. Whether these odors that contain many chemical compounds can be put in the type of the group of the 4 groups is not clear. With these unclear things included in the main part of the study that the authors are focusing, it is hard to interpret.
Comments by line numbers:
Line 78: “smell some common olfactory substances: vanillin, coffee, and/or lavender, menthol and/or ammonia, cinnamon” -> there seem no scientific rationales in the selection of the types of smells.
Line 81: “during quarantine” -> the duration of quarantine is different depending on the country and also depending on the timepoint during the pandemic. It is necessary to write how the quarantine duration was here.
Line 86-88: The reference cited here does not classify the odors to these 4 groups, and this is not right. If there are some other papers classifying odors in these 4 groups, you need to cite them. Besides, the paper cited, i.e., [14], is using 2 chemical compounds that activates only olfactory receptors and olfactory and trigeminal sense, respectively, and they are not classifying various chemical compounds into groups. If you confirmed that these odorous chemical compound and materials can be grouped in the groupings that you showed, it is necessary to cite that specific paper.
Also, coffee and lavender, cinnamon are not single chemical compounds. They contain hundreds of chemical compounds in their odors, and I don’t think that all these chemical compounds in the odors of coffee, lavender, and cinnamon can be classified in the same way. This is a big problem because the main results of the paper are based on the grouping and the differences in the recovery of sense depending on the type of odors.
The paper reports the results on hypo/ageusia but does not show if there were any correlation between the occurrence of taste dysfunction and smell dysfunction. Was there any correlation?
If possible, I recommend to reorganize the survey and redo the survey. If that is not possible, I hope the authors can reorganize the whole paper.
Author Response
Thanks for the suggestions. Comments by line numbers:
- Line 78: “smell some common olfactory substances: vanillin, coffee, and/or lavender, menthol and/or ammonia, cinnamon” -> there seem no scientific rationales in the selection of the types of smells.
Following your advice, we wrote the scientific name of the substances, and we added the reference to explain the choice of this odorants. Moreover, we corrected the table 1 and the figure 1 also.
- Line 81: “during quarantine” -> the duration of quarantine is different depending on the country and also depending on the timepoint during the pandemic. It is necessary to write how the quarantine duration was here.
Thank you for the suggestion. We added these data.
- Line 86-88: The reference cited here does not classify the odors to these 4 groups, and this is not right. If there are some other papers classifying odors in these 4 groups, you need to cite them. Besides, the paper cited, i.e., [14], is using 2 chemical compounds that activates only olfactory receptors and olfactory and trigeminal sense, respectively, and they are not classifying various chemical compounds into groups. If you confirmed that these odorous chemical compound and materials can be grouped in the groupings that you showed, it is necessary to cite that specific paper.
We are sorry for the mistake. The reference 14 was incorrect: we add the correct reference of the same author, and we corrected the substances following his classification.
- Also, coffee and lavender, cinnamon are not single chemical compounds. They contain hundreds of chemical compounds in their odors, and I don’t think that all these chemical compounds in the odors of coffee, lavender, and cinnamon can be classified in the same way. This is a big problem because the main results of the paper are based on the grouping and the differences in the recovery of sense depending on the type of odors.
Following the previous published articles, we correct the name of the substances: pure olfactory was phenyl ethyl alcohol, mixed olfactory / trigeminal substance was eucalyptol, mixed olfactory / trigeminal / gustatory substances was eugenol.
- The paper reports the results on hypo/ageusia but does not show if there were any correlation between the occurrence of taste dysfunction and smell dysfunction. Was there any correlation?
We couldn’t calculate the possible statistical correlation because we didn’t include patients without hypo/anosmia.
- If possible, I recommend to reorganize the survey and redo the survey. If that is not possible, I hope the authors can reorganize the whole paper.
Following your recommendations, we tried to modify the manuscript.
Reviewer 3 Report
Revisions:
- The title of the study does not correspond to the objectives of the study, since the objectives do not mention nerve pathways at any time. I suggest adapting the objective and/or the title of the study.
- According to the results and conclusions of the study, not only aspects of hypo/anosmia were investigated, but also those related to hypo/ageusia, so perhaps they should be included as a secondary objective of the study.
- In Table 2 I miss several post covid symptoms included in the questionnaire that was passed to patients such as the number of patients presenting with hyposmia or anosmia, as well as the separate presentation of patients presenting with hypogeusia or ageusia. I also miss the results regarding the average time it took the patients to recover each of the odors examined.
- Given that the results section (line 128-130) talks about the possible combinations in the recovery of perception of the substances tested, and that the aim of the study is to evaluate the order of recovery of the perception of certain odors, I think it would be more correct if the combinations in Table 3 and the sequences in Figure 5 were expressed in relation to the substances tested (vanilla, coffee/lavender, mint/ammonia and cinnamon) and not to the nerve pathways associated with them. This would make it easier to identify in the text and more in line with the objective of the study.
- Among the limitations of the study it is stated: "No distinction was made between hyposmia and anosmia symptoms, as they were considered as a single category". If no distinction was made between the two symptoms, why did the questionnaire include relative and differentiating questions for both symptoms?
Author Response
Thanks for your suggestions.
- The title of the study does not correspond to the objectives of the study, since the objectives do not mention nerve pathways at any time. I suggest adapting the objective and/or the title of the study.
Thank you. we modified the title writing “Hyposmia in Covid-19: temporal recovery of smell. A preliminary study.”
- According to the results and conclusions of the study, not only aspects of hypo/anosmia were investigated, but also those related to hypo/ageusia, so perhaps they should be included as a secondary objective of the study.
Following your advice, we added hypo/ageusia as secondary objective of the study.
- In Table 2 I miss several post covid symptoms included in the questionnaire that was passed to patients such as the number of patients presenting with hyposmia or anosmia, as well as the separate presentation of patients presenting with hypogeusia or ageusia. I also miss the results regarding the average time it took the patients to recover each of the odors examined.
We didn’t differentiate between hyposmia and anosmia or between hypogeusia or ageusia. We include a total of 181 patients with hyposmia or anosmia; 170 of 181 suffered from hypogeusia or ageusia too.
Moreover, we added the average temporal recovery of each odorant.
- Given that the results section (line 128-130) talks about the possible combinations in the recovery of perception of the substances tested, and that the aim of the study is to evaluate the order of recovery of the perception of certain odors, I think it would be more correct if the combinations in Table 3 and the sequences in Figure 5 were expressed in relation to the substances tested (vanilla, coffee/lavender, mint/ammonia and cinnamon) and not to the nerve pathways associated with them. This would make it easier to identify in the text and more in line with the objective of the study.
Each substances stimulates a different nerve pathway. However, following your suggestion, we modified the table 3 and figure 5, writing the smell substances instead of nerve pathways.
- Among the limitations of the study it is stated: "No distinction was made between hyposmia and anosmia symptoms, as they were considered as a single category". If no distinction was made between the two symptoms, why did the questionnaire include relative and differentiating questions for both symptoms?
It was a mistake. We corrected the questionnaire.